

# Self-aware face emotion accelerated recognition algorithm: a novel neural network acceleration algorithm of emotion recognition for international students

Lian Tong[1,*], Lan Yang[1,*], Xuan Wang[1] and Li Liu[2]

[1] Department of Computer Science and Engineering, Changsha University, Changsha, Hunan, China
[2] Department of Science and Engineering, Yamagata Univesity, Yonezawa, Yamagata, Japan
* These authors contributed equally to this work.

## ABSTRACT

With an increasing number of human-computer interaction application scenarios, researchers are looking for computers to recognize human emotions more accurately and efficiently. Such applications are desperately needed at universities, where people want to understand the students' psychology in real time to avoid catastrophes. This research proposed a self-aware face emotion accelerated recognition algorithm (*SFEARA*) that improves the efficiency of convolutional neural networks (CNNs) in the recognition of facial emotions. *SFEARA* will recognize that critical and non-critical regions of input data perform high-precision computation and convolutive low-precision computation during the inference process, and finally combine the results, which can help us get the emotional recognition model for international students. Based on a comparison of experimental data, the *SFEARA* algorithm has *1.3×* to *1.6×* higher computational efficiency and 30% to 40% lower energy consumption than conventional CNNs in emotion recognition applications, is better suited to the real-time scenario with more background information.

## INTRODUCTION

According to psychologist J. A. Russell, only 7% of the information in daily communication is conveyed *via* words, 38% *via* sounds, and 55% *via* facial expressions (*Mehrabian & Wiener, 1967*). This demonstrates that expressions can convey many subtle messages that words and sounds cannot, emphasizing the importance of expression information in human-to-human communication. Facial expression recognition has gradually become an important technology in interactive artificial intelligence as artificial intelligence and pattern recognition have advanced. Facial expression recognition technology is a component of emotional computing research, a difficult cross-disciplinary topic in physiology, psychology, image processing, pattern recognition, and machine vision. Facial expression recognition is the process of extracting features and classifying expression information from human faces, allowing computers to perceive human

Corresponding author
Li Liu, tl@ccsu.edu.cn

expression information and infer human psychological states, enabling intelligent interaction between humans and machines. This is an exciting aim, these topics have multiple application scenarios, such as human–computer interaction, intelligent control, security, medical, and communication scenarios.

Today's international students face increased social pressure (*Misra & Castillo, 2004*), primarily from their parents, their studies, and society. To relieve stress, many university students choose to express on social media (*Coates, Sykora & Jackson, 2019*). College students are eager to express themselves through social media platforms. Social media refers to websites and technologies that enable people to write, share, comment, discuss, and communicate (*Zobeidi, Komendantova & Yazdanpanah, 2021*). It can provide users with a lot of room for participation, and satisfy not only the requirement to store personal information but also the need to "be found" and "appreciated". Furthermore, the education by studying abroad, which is affected by the epidemic and fully implements long-distance teaching, and exchange, is facing unprecedented challenges, and such assistive technology is urgently needed to improve the quality of education management. Therefore, this research focuses on emotion recognition for international students, and this work will be useful for real-time tracking and early warning work for international students who exhibit abnormal emotions.

Our contributions are summarized as follows:

### Construct dataset

Existing emotion datasets are biased toward facial information, such as IAPS (*Lang, Bradley & Cuthbert, 2005*), FlickrLDL, and TwitterLDL (*Yang, Sun & Sun, 2017*), which generally only have head information, and lack the person's physical description and background information, making it difficult for the NN to respond to real emotional changes, and this type of data is scarce in reality. In this study, we created and annotated a dataset of 600 images of international students' emotions. We could achieve a more accurate emotion recognition model by responding to the emotional changes of international students more effectively using this dataset. The dataset includes comprises students from various countries who gave their permission for their social media data to be distorted and deformed to achieve diversity in the dataset.

### Portrait recognition algorithm selection

In this study, experiments were performed using several public datasets to assess the recognition abilities and performance of different convolutional neural networks (CNNs), and a selection was made to demonstrate the efficacy of our approach.

### Novel emotion recognition algorithm

In this study, a self-aware accelerated facial emotion recognition algorithm (SFEARA) is suggested. The algorithm increases the real-time performance of the CNNs in facial emotion recognition (FER), preprocesses the image, and uses hybrid accuracy for the feature extraction of different multiple regions to lower reduce the computation and energy consumption in convolutional operations.

## Related work

In this study, we focus on facial emotion recognition of international students, and through certain online social media image data of university students during their study period, our proposed algorithm can recognize their emotions faster and improve the real-time analysis capability of the entire emotion recognition system. In the existing public benchmarks, human expressions are generally classified into six categories, namely happy, anger, disgust, sad, surprise, fear, but there are new datasets released since 2020's that include compound expressions and thus reach 11 categories. Presently, the application scenarios of expression recognition are primarily human–computer interaction, precise direction of advertising, and student status recording. FER can be divided into two categories: traditional recognition methods (*Lyons et al., 1998*) and deep learning methods (*Zhang, Quan & Ren, 2016*), both of which can perform emotion recognition. However, the energy consumption of various processing methods varies. This section discusses the existing methods for sentiment recognition, including traditional methods and deep learning methods.

## Application of traditional emotion recognition in emotion recognition scenarios

Feature extraction in traditional emotion recognition relies heavily on mathematical methods and computer technology to organize and process data from digital images of facial expressions, extract expression features and remove non-expression noise. This feature extraction algorithm is for the image's main features, which objectively reduce the image's dimensionality, and therefore these feature extraction algorithms affect dimensionality reduction. At this point, the expression feature extraction algorithms are classified as static image-based feature extraction methods (*e.g.*, holistic and local methods) and dynamic image-based feature extraction methods (*e.g.*, optical flow, model, and geometric methods), with the distinction, made based on the state of the expressions and the object to be processed. Among them, static images show a single image of the expression when it occurs, whereas dynamic images show the movement of the expression between multiple images.

Several studies (*Wang, Huang & Wang, 2011*; *Zou, 2008*; *Ying & Cai, 2009*; *Ma & Khorasani, 2004*) have been conducted as holistic feature extraction methods using static images. The classical algorithms in holistic methods include PCA, independent component analysis (ICA), and linear discriminant analysis (LDA). Researchers have significantly contributed to this effort. *Wang, Huang & Wang (2011)*, *Zou (2008)* extracted expression features using the FastICA algorithm, which not only inherits the ICA algorithm's ability to extract hidden information between pixels but also can quickly complete the separation of expression features through iteration. *Ying & Cai (2009)* proposed a support vector discriminant analysis algorithm based on Fisher LDA and support vector machine that can maximize the inter-class separation of expression data with small sample data and does not require the decision function required to develop the support-vector machine (SVM) algorithm. *Ma & Khorasani (2004)* used the two which they then combined with neural networks to classify expression features. Because facial

expressions on still images have both overall variations and local variations, the information contained in local variations, such as texture and folds of facial muscles, can aid in accurately determining expression properties. Therefore, studies by *Kyperountas, Tefas & Pitas (2010)*, *Zheng et al. (2006)*, *Fu & Wei (2008)* examined the local information of still images. *Kyperountas, Tefas & Pitas (2010)* used various feature extraction algorithms such as Gabor and wavelets, as well as a new classifier for experiments on still images. In *Zheng et al. (2006)*, 34 face feature points are first manually labeled, then their the Gabor wavelet coefficients of the feature points are represented as marker map vectors, and finally the KCCA coefficients between the marker map vectors and the expression semantic vectors are calculated to achieve expressions classification.

*Yacoob & Davis (1996)* proposed using the optical flow field and gradient field between consecutive frames to represent the temporal and spatial variation of the image, respectively, to achieve the expression region tracking for each face image; then, the motion direction change of the feature region is used to represent the motion of facial muscles, which corresponds to different expressions. *Tsalakanidou & Malassiotis (2010)* proposed an ASM-based 3D face feature tracking method that tracks and models 81 face feature points to recognize certain composite action units. *Wang & Yin (2007)* proposed a method to recognize face movements and expressions using an image terrain feature model. The method is to procure face expression regions by tracking face feature points using AAM and manual annotation and then procure topographic features by computing the topographic histogram of face expression regions to achieve expression recognition. *Sung & Kim (2008)* proposed an AAM algorithm based on apparent 2D features and 3D shape features to achieve expression in an environment where the face position is shifted feature extraction. Other researchers have claimed that the generation and expression of expressions are heavily dependent on the changes in the facial organs that are represented. The major organs of the face and their folds are the regions where expression features are concentrated. Therefore, marking feature points on the facial organ regions and calculating the distance between feature points and the curvature of the curve where the feature points are situated becomes a method for extracting facial expressions using geometric forms. For example, the study by *Kotsia & Pitas (2007)* includes gridding the faces with different expressions using a deformation grid and using the change in grid node coordinates between the first and maximum frames of that expression sequence as geometric features for expression recognition.

The above methods are popular in emotion recognition scenarios, but because of their high computational effort and unsatisfactory accuracy, new solutions are urgently required to address the urgent need for emotion recognition scenarios. For example, wearing a face mask for social distancing related to COVID-19 reduces the recognition rate of facial expressions (*Pazhoohi, Forby & Kingstone, 2021*).

## Application of convolutional neural network in sentiment recognition scenarios

Deep learning differs from traditional methods, especially CNN has a strong feature extraction capability for images, which can avoid the complex algorithm of traditional

extraction feature methods and has higher recognition efficiency. Traditional face recognition extracts features from images through models, such as the commonly used 68 facial feature points, with low efficiency and accuracy, while deep learning can learn features more generally without a dedicated feature extraction step, greatly improving the efficiency and fault tolerance of recognition. Recently, using deep neural networks (DNNs) to solve FER problems has become prevalent because of the advancement of DNNs, which has aided in the advancement of sentiment analysis. As computational power increases, the structure of DNNs provides a good foundation for computer self-learning, allowing it to identify complex patterns in text, images, and sounds. Therefore, using DNNs to solve the FER problem is extremely appropriate. Among them, convolutional neural networks (CNNs) (*Zhang, Quan & Ren, 2016*) are popular in computer vision, and facial emotion recognition systems can be designed in a less time by feature extraction and classification during training, and such systems outperform traditional methods in real-time inference. With more research and practice, multiple network models such as AlexNet, GoogLeNet (Inception), VGGNet, and ResNet have performed well. These networks have been tested using a powerful dataset ImageNet, and therefore they are often used for image classification problems. For the construction of the network, the features of the image are usually extracted by a CNN layer, and then a fully connected layer performs nonlinear classification, similar to MLP (Multilayer perceptron), except those mechanisms, such as dropout, are added to avoid overfitting. In the last layer, different classifications correspond to different neural units, and the samples obtained from the variations of softmax belong to different types of probability distributions. *Miller & Fellbaum (1991)* proposed large datasets with 1 million images to annotate natural emotional facial expressions, which is quite applicable to computer vision, affective computing, social cognition, and neuroscience. *Corneanu, Madadi & Escalera (2018)* proposed a deep neural network architecture that solves both issues by combining them in the initial stage. *Wang et al. (2022)* proposed a new scheme for compressing datasets by aligning features (CAFE) that explicitly attempts to preserve the true feature distribution and the discriminative power of the resulting synthetic set, making it useful for various of architectures with a high degree of generality and improves performance by 11%. A very efficient CNN model optimization-based on AdaMax and ReLu algorithm (*Sezer & Altan, 2021*) was proposed to accurately identify faulty welds at the early stage. The robust hybrid classification model (*Yağ & Altan, 2022*) was designed with the flower pollination algorithm (FPA) and support vector machine (SVM) to accurately classify plant leaf diseases in real time, and got an 8.1% improvement in the same condition. The OCNN model (*Tang et al., 2023*) was improved by Octave convolution algorithm and DyReLU function, obtained a higher accuracy and little lower computational consumption.

Table 1 compares the results of the overall approach, the local approach and the CNN for image accuracy (acc) and single image running time (sirt) in sentiment process recognition for a single image.

Generally, recognizing sentiment using deep learning is the most suitable approach, but the real difficulty is caused by the performance of processing applications. Although these methods are extensively used in industry and academia, there are few studies that balance

**Table 1 Comparison of the effect of traditional recognition methods and convolutional neural network recognition.**

|  | acc | sirt |
|---|---|---|
| Holistic approach (TFFM) | 55.3% | 524 ms |
| Local method (Gabor) | 65.6% | 354 ms |
| CNN (AlexNet) | 85.2% | 36 ms |

accuracy and efficiency, or greatly improve efficiency, especially in the field of education management of international students. Regarding the question "how to process input data quickly and obtain better results in the FER domain", in this study, a novel neural network acceleration algorithm of emotion recognition for international students is proposed to respond to improve the quality of education management, and has achieved a significant improvement in efficiency with almost no decrease in accuracy and more complete metrics, such as single image runtime (sirt), average mask (am), and the sum of running time on the whole dataset (SumRuntime). This may indeed help universities to understand the psychological conditions of students and take timely action, and also lower the applicable threshold.

### CNN overview

CNN has two parts; they are feature extraction and full connection classification. Figure 1 describes the forward processing of CNNs Feature extraction belongs to the convolutional layer while full connection classification belongs to the full connection layer (*Goodfellow et al., 2016*; *Gu et al., 2018*). In the convolutional layer, it analyzes the input data set in batches and obtains different abstract features. The input includes $H$, $W$, and channel ($H$ and $W$ are the image resolution ratio while the channel is the number of channels), *e.g.*, $r$, $g$, $b$. Assuming that the convolutional layer uses the filters of dimension ($Df{:}Hf{:}Wf$) to extract feature map $a$, $a_{i{:}j}$ is the value of the current feature map in row $i$ and column $j$, as shown in Eq. (1):

$$a_{i,j} = f\left(\sum_{d=1}^{D_f}\sum_{m=1}^{H_f}\sum_{n=1}^{W_f}\left(w_{d,m,n} \times x_{d,i,j}\right) + w_b\right). \tag{1}$$

$D_f$ stands for depth, $H_f$ stands for height, $W_f$ stands for width, and $f$ is an activation function such as sigmoid, hyperbolic tangent (tanh), or rectified linear units (ReLU). Training primarily includes forward propagation, backward propagation, and kernel update. The inference process is forward propagation only. The network model is generated by the training of CNN, which requires to be fine-tuned and optimized before starting to perform inference on different data sets and then completing the inference.

### Forward propagation

The forward propagation algorithm of DNN (*Alpaydin, 2016*) is to use certain kernel coefficient matrix $W$ and bias vector $b$ and input vector $x$ to multiple linear and activation

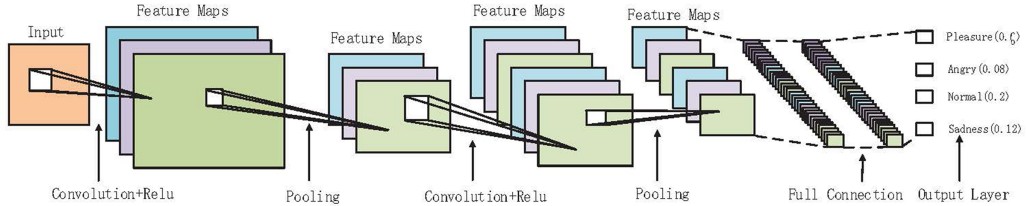

**Figure 1 In-depth description of the forward propagation process of convolutional neural networks with the final classification.**

function operations. We will obtain a result from the input layer and calculate layer by layer till output layer.

Assuming there are m nerve cells in $l-1$, then as to the output $a_j^l$ of nerve $j$ in layer $l$, we can obtain Eq. (2):

$$a_j^l = \sigma\left(z_j^l\right) = \sigma\left(\sum_{k=1}^m w_{jk}^l a_k^{l-1} + b_j^l\right). \tag{2}$$

Generally, Input: total numbers of layers $ls$, all the corresponding matrixes of hidden layers and output layer, offset vector $b$, bias vector $b$, input vector $x$. Output: Output $a^l$ in the output layer

1) Initialization $a^1 = x$;
2) for $l = 2$ to $ls$, calculate Eq. (3).

$$a^l = \sigma\left(z^l\right) = \sigma\left(W^l a^{l-1} + b^l\right). \tag{3}$$

Backward propagation is the process, in which the loss function of DNN uses GD (Gradient Descent), to obtain the iterative optimal and minimum value. Moreover, the loss function was used to measure the loss between the training samples and real training samples. The output of the training samples is to obtain $a^l$ in layer $l$ *via* forward propagation; therefore, it is the output from the forward propagation algorithm. In this passage, we use the mean squared error to measure the loss. In every sample, we expect a minimized Eq. (4):

$$J(W, b, x, y) = \frac{1}{2}\left|\left|a^l - y\right|\right|_2^2. \tag{4}$$

$a^l$ and $y$ are the vectors of the characteristic dimension, which is nout, $||S||_2$ is the L2 norm of $S$.

The backward propagation of DNN is listed as follows. Batch (BGD), small batch (MBGD) and random (SGD) are the three methods of gradient descent. If the sample size is relatively small, the batch gradient descent algorithm is used. If the sample is extremely large, or online algorithms, the stochastic gradient descent algorithm is used. In the practical general case, the small batch gradient descent algorithm is used.

Kernel update: kernel amendment is a process that amends network parameters *via* the calculated error loss. The purpose is to gradually change and decrease the output result and the stabilization of the results has leveled off. Moreover, the change in loss gets smaller.

Kernel equation: $W^+ = W - \eta \frac{\partial Loss}{\partial w}$. W is the kernel parameter, Loss is the loss result, $\eta$ is the learning rate. The closer it gets to the real result, the smaller changes that reciprocal of Loss will have. The gap between $W^+$ and $W$ was smaller. It has a geometric significance, as shown below:

When point *P* wants to get to the lowest point, *P* must move left on the *x*-axis, which means adding a negative number. Now that the gradient is positive, it has to be minus (learning rate $\times$ partial derivative).

When point *P* is on the left of the lowest point, if it wants to get to the lowest point, *P* must move right on the *x*-axis, which means adding a positive number. The gradient is negative, and so it has to be minus (learning rate $\times$ partial derivative).

Through the derivation, we can identify the direction of point *x*'s movement (derivative). Then, we set the distance in every move (learning rate $\eta$) to avoid the optimum local situation. The learning rate should be adjusted as per the real situation. Once confirming the two factors, we could identify the lowest point. As long as there are no obvious loss changes during the training process, we could ensure that the training parameter results are stable, and the model could be used to for inference.

Generally, both the training and inference processes of CNN are extremely important in image recognition. The training of CNN provides the image recognition model, while the inference is inferred from the model by forward computation.

## METHOD

In this section, we describe the international student emotion dataset constructed in this article, and propose a SFEARA in an emotion recognition scenario. The primary scheme is as follows. First, we describe the data types used in training and testing; next, we discuss the selection strategy in the face recognition algorithm; then, SFEARA is proposed based on the selection results of the face recognition algorithm.

### Construct dataset

In this study, we used publicly available image sentiment datasets, including MIRFLICKR-25K (*Huiskes & Lew, 2008*), MIRFLICKR-1M (*Huiskes, Thomee & Lew, 2010*), and Fer2013 (*Goodfellow et al., 2015*). Images containing human faces were selected from these three datasets using random sampling, and emotions were classified as pleasant, normal, sad, and angry. The MIRFLICKR-25K dataset is a collection of 25,000 images downloaded from the public API of the social photography site Flickr, plus full human annotations. MIRFLICKR-1M was obtained using the same data collection method as MIRFLICKR-25K, but without manual annotation of all images, all of which are available under a Creative Commons Attribution License. The Fer2013 dataset is derived from the Kaggle Facial Expression Contest dataset and comprises seven categories of emotional images. We randomly selected 1,269 portrait images only with one of facial sentiment in pleasure, normal, sadness, and anger from MIRFLICKR-25K, of which 80% of the images were

divided as training set and 20% as the test set. A total of 23,308 portrait images only with one of facial sentiment in pleasure, normal, sadness, and anger were randomly selected from the MIRFLICKR-1M dataset, of which 80% of the images were used as the training set, and 20% of the images were divided as the test set. They are checked and annotated by hand, also compared the features extracted from two datasets. If the contrast ratio exceeds 99%, it is considered the same image and an image is deleted. We just selected the images which have only one emotion to guarantee the images have the same annotations as Fer2013. The Fer2013 dataset comprises 35,886 images of faces with different expressions, of which there are 28,708 images in the training set and 3,589 images in each of the validation and test sets. We used the dataset's test set to compare the inference results' accuracy and efficiency. To study the required consistency, in this study, the three datasets were uniformly reclassified, grouping the emotion categories into pleasure, normal, sadness, and anger. We use the unified classification data of these three recognized categories for model training in preparation for the real-time inference session.

To some extent, text data on social media can accurately reflect college students' emotional states. There are various types of social media, and this study examines "Twitter" and "Facebook", the mainstream media society as well as face emotion recognition (FER). In addition to the above public dataset, 150 portrait images posted on Twitter and Facebook by college students from different countries (Japan/Taiwan/India/English/South Africa/America, *etc.*) were collected in this study to obtain a more suitable model for international students. These images are expanded from 150 to 600 images by image processing such as perspective transformation (*Mezirow, 1978*), affine transformation (*Modenov & Parkhomenko, 1965*) and Gaussian blur (*Hummel, Kimia & Zucker, 1987*). The expansion in this manner improves the robustness of the model trained by CNNs. To improve the classification accuracy, images in social media are generally equipped with text, and we use the SVM vector machine algorithm to classify the text in the images, and the text corresponds to the corresponding category of the images. This dataset can help us develop a robust emotional inference model for international students, which can be used to study the emotional changes of college students during their life and study. In this study, to annotate the dataset more accurately, the SVM vector machine algorithm is used to pre-annotate the emotion of the text in the life status posted by international students, and then the corresponding images are manually proofread to achieve accurate annotation of the ISSM DataSet. The generalizability of this study is confirmed by fine-tuning the trained model using ISSM DataSet and employing inference operations. For this purpose, we use 70% of the images in the ISSM DataSet as the fine-tuned training dataset and 30% as the inference dataset. The classification of images in each dataset is shown in Table 2.

## Preprocessing algorithm selection

In this study, we focus on the emotion recognition of international students; to be able to improve the real-time emotion detection, we use a face detection algorithm to preprocess the data and improve the efficiency of emotion recognition by masking during fine-tuning and inference, and the accuracy will only be reduced within 3.3%. To suppress the

| DataSet | Pleasure | Normal | Sadness | Anger | Sum |
|---|---|---|---|---|---|
| MIRFLICKR-25K | 539 | 230 | 264 | 236 | 1,269 |
| MIRFLICKR-1M | 7,905 | 8,525 | 3,954 | 2,923 | 23,307 |
| Fer2013 | 8,308 | 3,972 | 3,864 | 3,544 | 19,688 |
| ISSM_DataSet | 124 | 134 | 94 | 248 | 600 |

**Table 2 Image sentiment data set.**

uncertainties efficiently and prevent deep networks from over-fitting uncertain facial images, *Wang et al. (2020)* proposed a simple yet efficient SelfCure Network (SCN). We analyzed the most popular face recognition algorithms. Faster R-CNN, SSD-300, and multi-task convolutional neural network (MTCNN), and selected SSD-300 based on the results of accuracy and single image processing efficiency.

Faster R-CNN (*Ren et al., 2017*) integrates feature extraction, suggestion extraction, rect refine, and classification in a single network, allowing for comprehensive performance improvements, particularly in detection speed. First, Faster R-CNN uses a set of basic conv +relu+pooling layers to extract the feature maps of the image. Second, the RPN network is used to generate region proposals. This layer determines whether the anchor point is positive or negative by softmax, and then uses bounding box regression to correct the anchor point to obtain an accurate positive point. Third, this layer collects the input feature maps and proposals, combines this information to extract the proposed feature maps, and sends them to the subsequent fully connected layers to determine the target class. Finally, box regression, obtain the final accurate location of the detection box.

Single Shot MultiBox Detector (SSD) (*Liu et al., 2015*) removes the bounding box proposal and the subsequent pixel/feature resampling, which is considerably faster than Faster R-CNN and has higher detection accuracy than Faster R-CNN in terms of prediction on feature maps of different scales. The core of SSD prediction first determines the prediction box category (the one with the highest confidence) and confidence value, and filters out the prediction boxes belonging to the background and those with a low confidence threshold; then encodes the remaining prediction boxes to get the real position parameters (after decoding, it requires to clip to prevent the prediction box position from exceeding the decoding, the prediction boxes are sorted in the descending order as per the confidence level, and the top-k prediction boxes are retained. Finally, the non-maximum suppression algorithm is performed to filter out those prediction boxes with large overlap, and the remaining prediction boxes are the results.

Multi-task convolutional neural network (MTCNN) (*Zhang et al., 2016*) is used to balance performance and accuracy, avoiding the huge performance consumption associated with traditional ideas such as sliding window plus classifier. First, the test images are successively resized to obtain image pyramids. Second, a small model (P-Net) is used to generate target region candidate frames with certain probability. Third, a more complex model (R-Net) is used to finely classify and regress the frames in the region, and let this

**Table 3 Performance of different networks in different data sets.**

|  |  | Faster R-CNN | SSD-300 | MTCNN |
|---|---|---|---|---|
| MIRFLICKR-25K | acc (%) | 71.5 | 74.5 | 86.1 |
|  | sirt (ms) | 0.134 | 0.089 | 0.265 |
|  | am (%) | 76.2 | 83.1 | 65.2 |
| MIRFLICKR-1M | acc (%) | 79.5 | 86.2 | 88.6 |
|  | sirt (ms) | 0.17 | 0.075 | 0.34 |
|  | am (%) | 85.2 | 88.2 | 76.5 |
| Fer2013 | acc (%) | 64.2 | 69.5 | 81.5 |
|  | sirt (ms) | 0.054 | 0.031 | 0.24 |
|  | am (%) | 86.8 | 88.2 | 84.2 |

step proceed recursively. Finally, many candidate images are filtered out by R-Net and the results are input to (O-Net) to get the exact frame coordinates and landmark coordinates.

In this study, we measured the accuracy (acc), single image runtime (sirt), and average mask (am) of three face recognition networks in different datasets. Based on the experimental results of face detection accuracy, running time, and average mask as shown in Table 3, we selected the SSD recognition network with good accuracy and highest operational efficiency as the face preprocessing algorithm for SFEARA emotion recognition. Considering the effect of SSD preprocessing algorithm on overall performance, we selected the *confidence* as the key parameter and evaluated the best setting on our dataset, although we used the default parameters of SSD. The result showed that the *confidence* parameter setting in the 0.001–0.011 balances performance and precision best as Fig. 2, and its default value is 0.001.

## Design of SFEARA

The traditional emotion recognition methods and CNNs methods at the present stage are inefficient in the face of high real-time inference scenarios and cannot adapt to the urgent requirements of the day. To improve the speed of facial emotion recognition, this article proposes a SFEARA. Figure 3 depicts the use of the SFEARA method in the inference process. The algorithm has three advantages:

1) the input information can be preprocessed and decoupled using edge devices;
2) the local feature library is less noisy, and independent local is more conducive to global learning;
3) local learning is adaptive at different stages, and local learning is actually more appropriate at later stages.

The algorithm comprises three main parts. First, we preprocess the image with SSD, whose process is shown in Fig. 4, which includes convolution operations, determination of prediction frames, and classification. Then, combine the face detector api in OpenCV (*Nasrollahi & Moeslund, 2013*) and the model provided by the model in *Ren et al. (2014)* to

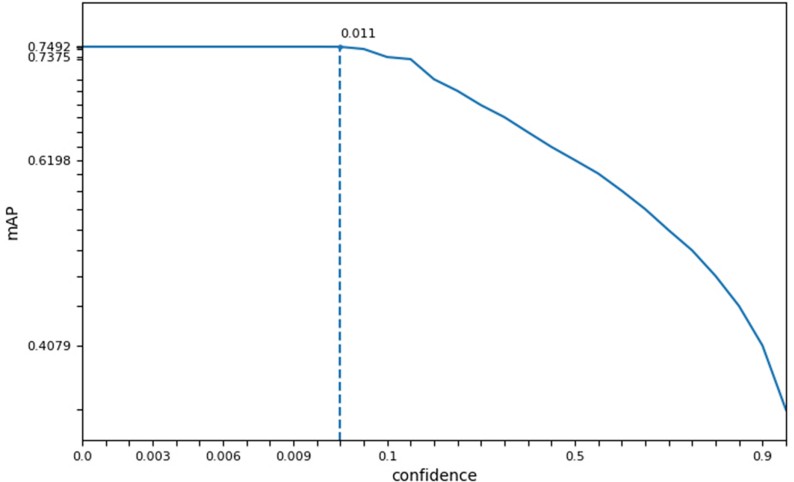

**Figure 2  The best *confidence* parameter of SSD on our dataset.**

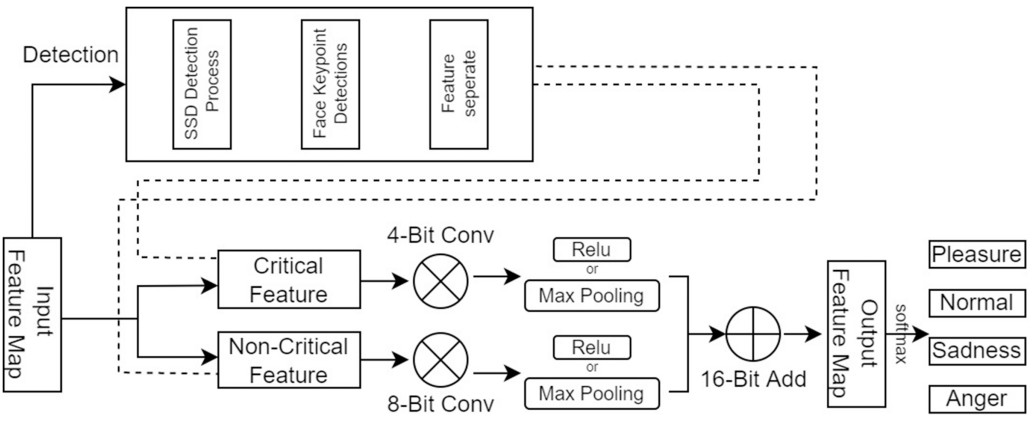

**Figure 3  SFEARA in the inference.**

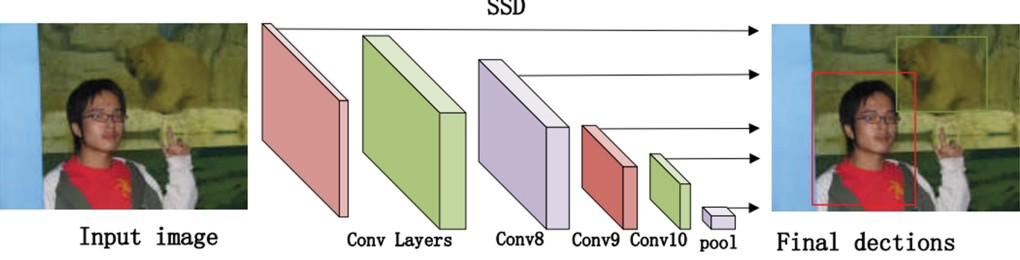

**Figure 4  The SSD-based object detection process.**

detect key points on the face and fuse the separated image data in a way as shown in Fig. 5. The approach is to detect key points of faces, filter non-face information, and qualify object frames twice, and by fusing in non-key regions, the result is more accurate information in face detection images. Next, a quadratic regression method based on LBF (local binary features) is proposed in *Ren et al. (2014)* to solve the face alignment problem; this method

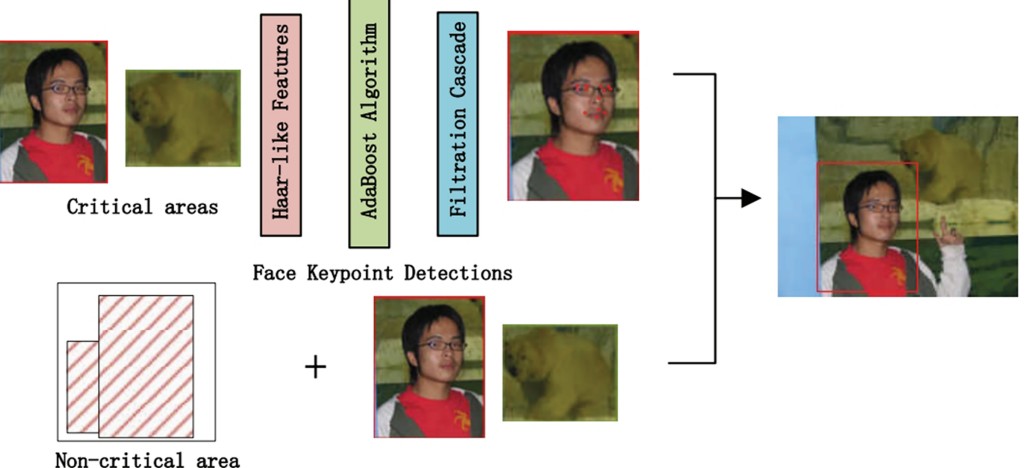

**Figure 5  Image data fusion and separation process.**

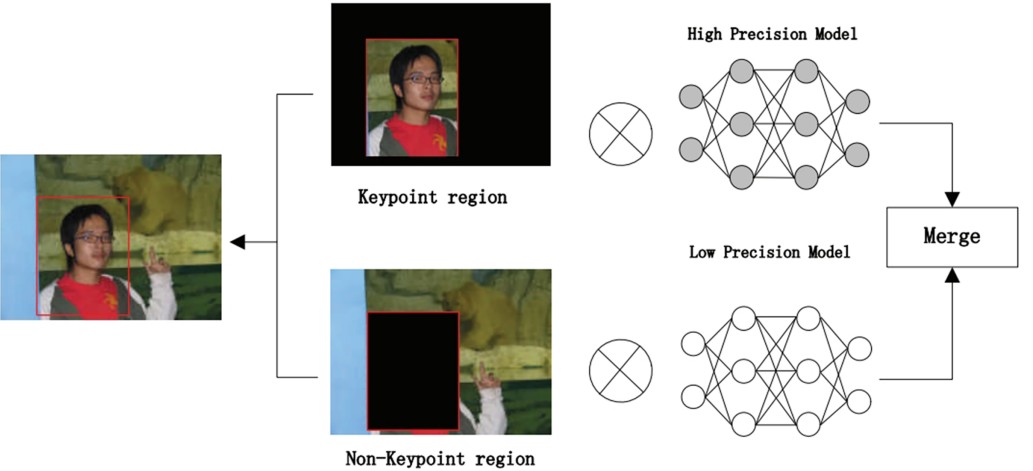

**Figure 6  SFEARA hybrid accuracy process.**

is based on the idea of local first and then whole. Finally, using the key point information identified by FacemarkLBF, after dividing the key and non-key regions, the CNN is finally used for inference classification, as shown in Fig. 6. The specific steps described by Algorithm 1.

As shown in Fig. 4, we use the object region coordinates obtained by SSD on the edge device and use the object region coordinates and the image to be predicted as the input to the algorithm in this study. Using the region information of object detection for face key point detection is an important of preprocessing step.

As shown in Fig. 5, we use the loaded face detector method to scan the object detection region data and determine the key points of faces. Because all face keypoint detection algorithms are a truncated face image, here we require to use the Haar face detector in OpenCV (*Nasrollahi & Moeslund, 2013*), and the program flow of process is as follows.

**Algorithm 1** Description of algorithm for key steps of emotion recognition.

**Input:** KeyAreaPointsSet:*RPSet*, Images: *Img*

**Output:** ClassificationResult:*Class_result*

1: **function** RECOGNIZEFACE(*Img*)
2:   Extract Haar-like Features from *Img*
3:   Classification according to AdaBoost Algorithm
4:   *FaceFlag* ← Filter by Cascade
5:   **return** *FaceFlag*
6: **end function**

7: **function** EMOTIONDETECTION(*RPSet,Img*)
8:   **for** *index, (xmin,ymin,xmax,ymax)* in enumerate(*RPSet*) **do**
9:    *face_Img* Depends on (xmin,ymin,xmax,ymax) get pixels from *Img*
10:    Construct FaceImg.append(*face_Img*)
11:   **end for**
12:   **for** *index, faceimg* in enumerate(*FaceImg*) **do**
13:    *FaceFlag* ← *RecognizeFace(faceimg)*
14:    **if** *FaceFlag*! = *True* **then**
15:     *RPSet.remove(index)*
16:    **end if**
17:   **end for**
18:   Construct *Mask* which *Mask.shape* == *Img.shape* and initial value is 1
19:   **while** *(xmin,ymin,xmax,ymax)* ← *RPSet* **do**
20:    set *Mask(xmin : xmax,ymin : ymax)* = 0
21:   **end while**
22: **end function**

23: **function** FORWARDCONV (*Img,Mask*)
24:   Conditional multiple convolution computation
25:   **for** *index,value* in enumerate(Mask) **do**
26:    **if** *value* == 1 **then**
27:     *ConvResult* ← Convolution calculation using low precision
28:    **else**
29:     *ConvResult* ← Convolution calculation using high precision
30:    **end if**
31:    *result* ← Concat *ConvResult*
32:   **end for**
33:   **return** *result*
34: **end function**

1) Use Haar-like features for detection.
2) Use an integral map to accelerate the solution of Haar-like feature values.
3) Use AdaBoost algorithm as a solid classifier for distinguish between human and non-human faces.
4) Cascade the strong classifier using filter cascade to improve the accuracy.

Then, avoid memory leaks by creating FaceMark objects using smart pointers (PTR). The face detector is then trained by loading a 300-W dataset. Each frame is run on the face detector to obtain the coordinates of the important regions of the face, and multiple rectangular containers are developed by limiting the coordinates of the key regions with a pixel threshold, whose main purpose is to limit the number of pixels in each container. The more containers there are, the fewer pixels are contained in a single container. Finally, the face information contained in the containers is used as the keypoint region. The keypoints are collected with different precisions to use them as input in subsequent inference. The keypoint regions inside the container are obtained by threshold restriction, and those outside the container are non-keypoint regions.

As shown in Fig. 6, we use CNNs to extract features from the preprocessing output, where the key regions of the features are inferred using a high precision model and the non-key regions of the features are inferred using a low progress model; Then, the results are fused and the prediction results are classified.

In forward propagation, the first layer of convolutional operations is computed with different accuracies based on the coordinates of critical and non-critical regions obtained using the above mentioned algorithm. When the region involved in the calculation is non-critical, we use low precision calculation, and when the region involved in the calculation is a critical region, a high precision calculation is used to calculate it. Based on the feature fusion of the convolution calculation, it is easy to determine the coordinates of the critical and non-critical regions after convolution. The tracking of the entire forward propagation of the convolution calculation can be achieved. The primary operation is to track the steps in the convolution process. The convolution kernel runs to the sensitive region, where we quantize the input features, converting from FP32 to INT8, and to the non-sensitive region, where we quantize the input features, converting from FP32 to INT4. This is a typical step in traditional quantization schemes because INT8 is sufficient for most inference workloads. Each time before computing the matrix-vector multiplier (MVM: matrix-vertex multiplier), the switch is made based on the coordinates of critical and non-critical regions. The computational accuracy is switched between critical and non-critical regions. When the coordinates of the filter run are in the critical region, we use int8 to represent the weight and feature values. The 8-bit convolution operation is performed in this region. When the coordinates of the filter run are in the non-critical region, we use int4 to represent the weight and eigenvalues and perform 4-bit convolution operations in this region.

Obviously, the computational consumption of the 4-bit operation is smaller than the 8-bit operation; this approach can effectively reduce the computation and improve inference performance. However, when the filter runs on the boundary of the mixed region, *i.e.*, the

critical region and the non-critical region, to ensure the feature information's validity, we all use the int8 method for convolutional operation. This method can guarantee the accuracy of the NN, but we require to retrain the network model and fine-tune the weights such that it can be more adaptable to this bit change and reduce the impact on the accuracy of emotion recognition of faces.

Using the SFEARA algorithm, in this study, we focus on the person's information using high precision and extract the background information using low precision. In this manner, people's emotions in images can be recognized more quickly, and the computational effort and energy consumption in the emotion recognition process are reduced.

## EXPERIMENT

All experimental environments in this study were conducted on a GeForce GTX 1080, 8 GB of video memory, TensorFlow 1.8.0, and PYTHON 3.6.3. This section demonstrates the acceleration performance and accuracy of the algorithm, and multiple sets of experiments are conducted to demonstrate the algorithm's effectiveness. The percentage of critical and non-critical regions are analyzed, and the reduced computation and memory access consumption are evaluated using simulation experiments.

We generated pre-trained models for all four neural networks (ResNet, AlexNet, GoogLeNet, VGGNet) by doing pre-training on four datasets (MIRFLICKR-25K, MIRFLICKR-1M, Fer2013 and ISSM DataSet). Each original model was fine-tuned using the ISSM DataSet training set to obtain a model more suitable for the emotion recognition of international university students. Table 4 compares the improvement results in ResNet, AlexNet, GoogLeNet, VGGNet using SFEARA method. The effect of the SFEARA method on the NN prediction with a roc curve is described, in Fig. 7.

Figure 8 shows the performance results of the CNNs after using the fine-tuned models compared with the CNNs added the SFEARA optimization scheme on the test set of the public datasets, where the left indicates the SFEARA method added and the middle indicates the original convolutional neural network method and the right indicates the OCNN method (*Tang et al., 2023*). The acc means accuracy, sirt means single image runtime, ec means energy consumption. We used the simulation method proposed in E2CNNs (*Ponzina et al., 2021*) to estimate the energy consumption. We performed a 100-fold expansion operation on the data of sirt to make the data results more elegantly displayed. Then, the inference process of AlexNet, VGGNet, GoogLeNet, and ResNet was optimized using SFEARA. The ISSM_DataSet test set was identified for inference and compared with the same network without SFEARA. The running time of single image, the running time of the whole data set, inference accuracy and energy consumption are primarily measured. For display convenience, we expanded sirt and sumtime by a factor of 100 and reduced them by 100, respectively. The experimental comparison results are shown in Fig. 9.

To describe the details of SFEARA method, we evaluated the percentage of critical and non-critical regions as Table 5 for MIRFLICKR-25K, MIRFLICKR-1M, Fer2013, and

**Table 4 Computation and memory access statistics for different data sets.**

| | Baseline | | Based on SFEARA | |
|---|---|---|---|---|
| | Computation (*10,000,000) | Memory access (*1,000) | Computation (*10,000,000) | Memory access (*1,000) |
| MIRFLICKR-25K | 195 | 30 | 136.305 | 10.485 |
| MIRFLICKR-1M | 195 | 30 | 111.93 | 8.61 |
| Fer2013 | 195 | 30 | 142.545 | 10.965 |
| ISSM_DataSet | 195 | 30 | 120.51 | 9.26 |

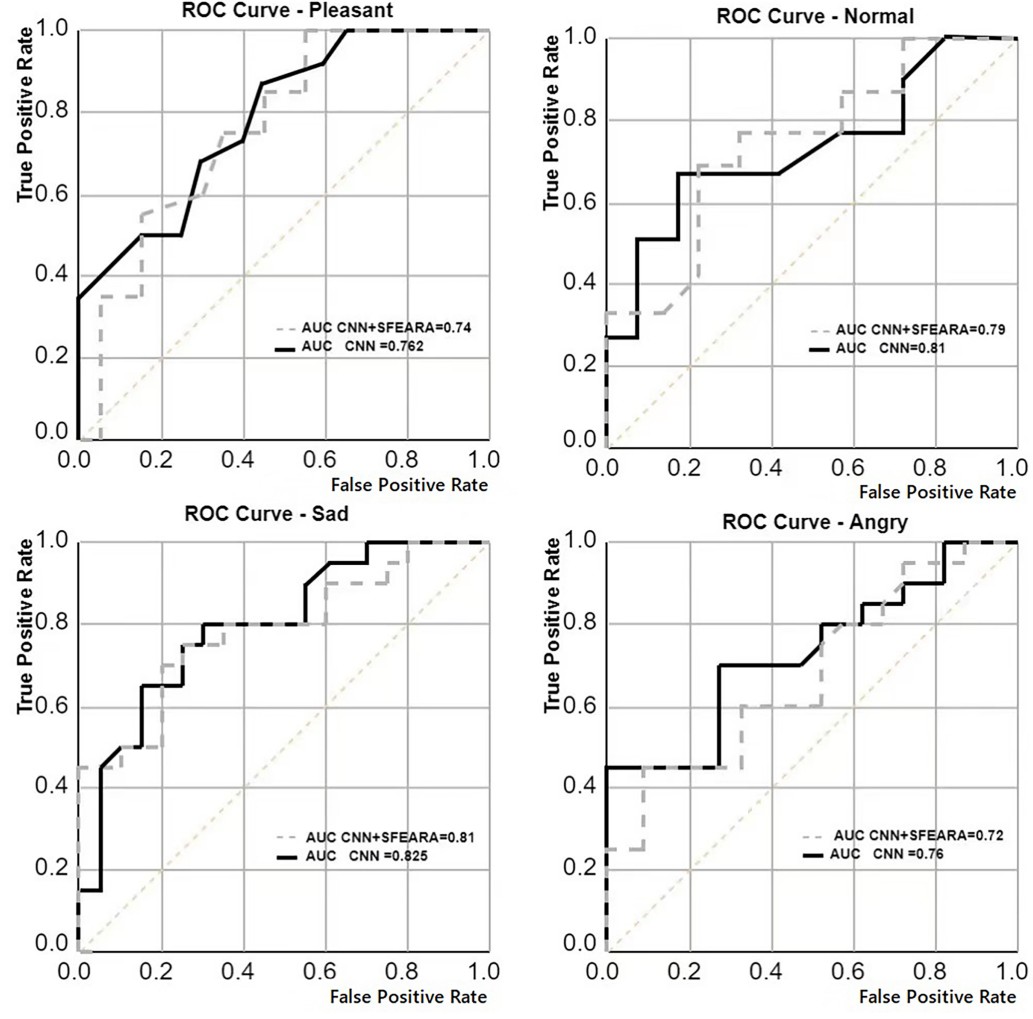

**Figure 7 Average comparison of the results of multiple CNNs (AlexNet, ResNet, GoogLeNet, VGGNet) with the average results of CNNs using the SFEARA method.**

ISSM DataSet test datasets. The reduced computation and memory access are estimated by simulation and based on the simulations of the AlexNet. As shown in Table 6, The Computation is the computation of the convolutional network and the Memory Access is

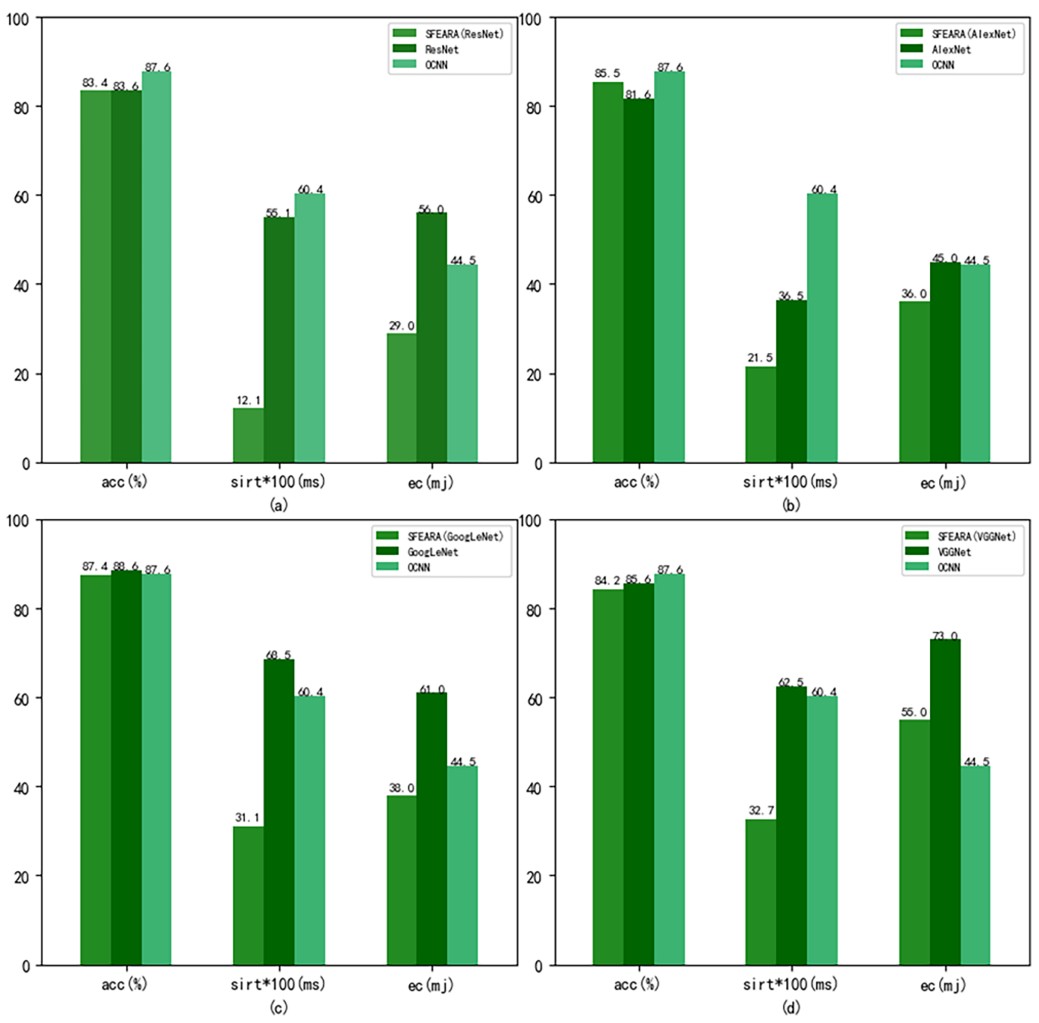

**Figure 8 Comparison of CNNs' performance with and without SFEARA and OCNN based on the public datasets.**

input memory access times of the convolutional network. They are compared with and without SFEARA optimization.

## DISCUSSION

Our aim in this study is to improve the efficiency of CNN-based FER in optimizing the time and energy consumption of the recognition process, especially for applying emotion recognition to college students from different countries. Table 4 and Fig. 6 show that our proposed method has a smaller impact (within 3.3%) on the accuracy of sentiment recognition, primarily because of the mixed precision processing introduced by the SFEARA method. At the same time, we reduced the computational consumption and memory access consumption by preprocessing the image to be inferred, determining the division of key regions, increasing the sparsity of the image itself, and simultaneously inferring the features in different regions using mixed precision as per the idea of mixed precision proposed by *Song et al. (2020)*. As shown in Fig. 7, It is experimentally

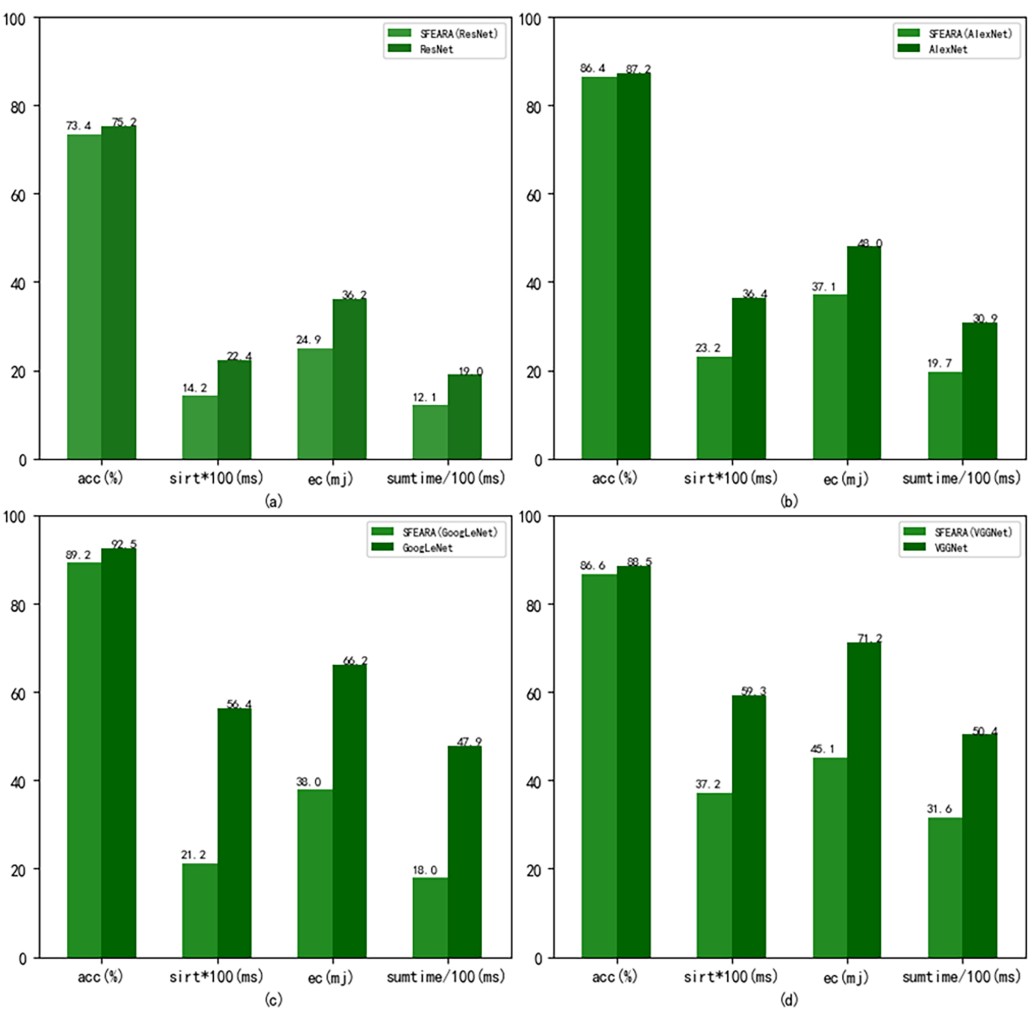

**Figure 9 Comparison of CNNs' performance with and without SFEARA including sumtime based on ISSM_DataSet.**

demonstrated that the algorithm yields better results in the accuracy (acc), the single image runtime (sirt) and energy consumption (ec) compared to the same type of unimproved network, even the latest research improved by Octave convolution algorithm and DyReLU function and named OCNN model. The results show that the performance of the SFEARA algorithm proposed in this study is improved for different CNNs with a speedup ratio of 1.3×–1.6×, while only within 3.3% reduces the accuracy and the energy consumption (ec) is reduced by 30–40%. With SFEARA, it is possible to identify critical and non-critical regions during inference dynamically, and it is friendly to use low precision optimization for non-critical regions. This is because it is not the key information that the recognition model focuses on. In this study, we take advantage of the low precision to add optimization operations to the CNNs to reduce the computation of GPU and reduce memory access consumption. Because this study is improved from the input perspective, and the model's accuracy is dynamically adjusted and more flexible, the existing method cannot be applied to the SFERRA algorithm. compared to the pure hybrid accuracy processing. By these

**Table 5 Performance of different networks in different data sets.**

| Emotion | Baseline | | | Based on SFEARA | | |
|---|---|---|---|---|---|---|
| | Precision (%) | Recall (%) | F1 (%) | Precision (%) | Recall (%) | F1 (%) |
| | | | ResNet | | | |
| Pleasant | 64.29 | 74.67 | 72.00 | 62.36 | 73.88 | 72.1 |
| Normal | 60.74 | 69.00 | 65.19 | 59.83 | 65.61 | 63.12 |
| Sad | 62.22 | 79.66 | 75.00 | 61.60 | 75.83 | 69.68 |
| Angry | 71.00 | 86.00 | 81.82 | 69.37 | 84.20 | 80.56 |
| | | | AlexNet | | | |
| Pleasant | 72.57 | 84.31 | 78.85 | 72.28 | 81.53 | 76.62 |
| Normal | 70.67 | 79.57 | 74.13 | 69.81 | 76.53 | 71.33 |
| Sad | 78.59 | 88.76 | 81.19 | 77.78 | 87.72 | 83.79 |
| Angry | 75.44 | 87.33 | 81.76 | 74.53 | 85.11 | 80.32 |
| | | | GoogLeNet | | | |
| Pleasant | 74.70 | 86.67 | 83.02 | 74.10 | 83.25 | 80.36 |
| Normal | 75.76 | 89.29 | 81.97 | 75.38 | 87.22 | 82.63 |
| Sad | 84.10 | 93.10 | 90.00 | 83.51 | 89.68 | 87.19 |
| Angry | 88.76 | 86.00 | 81.36 | 88.67 | 80.65 | 85.52 |
| | | | VGGNet | | | |
| Pleasant | 75.53 | 85.33 | 80.12 | 74.09 | 82.15 | 78.24 |
| Normal | 77.63 | 89.29 | 82.65 | 76.23 | 86.36 | 81.25 |
| Sad | 89.67 | 92.66 | 91.14 | 87.97 | 90.69 | 90.21 |
| Angry | 87.55 | 83.10 | 84.76 | 86.32 | 80.31 | 84.69 |

**Table 6 Proportional distribution of critical and non-critical areas.**

| | Qty | Critical areas | Non-critical areas |
|---|---|---|---|
| MIRFLICKR-25K | 1,269 | 30.1% | 69.9% |
| MIRFLICKR-1M | 23,307 | 43.6% | 57.4% |
| Fer2013 | 19,688 | 74.1% | 26.9% |
| ISSM_DataSet | 600 | 38.2% | 61.8% |

means, the performance of the inference process can be effectively improved. Therefore, the algorithm proposed in this study is effective for different NNs and outperforms the unoptimized CNNs.

The above experimental results show that adding SFEARA optimization scheme to the inference process can reduce the computation and memory access consumption of the inference process. The images in MIRFLICKR-25K, MIRFLICKR-1M, and ISSM DataSet are primarily certain life photos containing people information and a lot of background information, so the performance improvement is more. However, Fer2013 is mainly portrait information and less background information, so this method improves the performance less. Therefore, this method is more suitable for images with more

background information and has limited performance improvement in the face of images with less background information, which is a limitation of the SFEARA algorithm; however, from the overall network optimization level, this method can serve both cases, and only the acceleration in the face of the second case is less effective.

We require to state that the emotional problems of international students are a hot issue of great concern to all countries, and these people carry the country's hope and future, and how to appropriately respond to the behavior of negative emotions of university students is a problem that every university requires to face. Therefore, it is essential to effectively identify the emotional changes of international students effectively. The preprocessing scheme used in this study will dynamically identify faces, retaining the main information of emotions and weakening non-critical information. Therefore, it is particularly suitable for this study to apply the double precision optimization scheme in training and inference. The combination of fine-tuning, sparsification, and computational accuracy substantially improves the efficiency of training and inference with little impact on accuracy.

## CONCLUSION

In this study, we propose a novel emotion recognition acceleration algorithm based on international students. It is well known that using sufficient datasets in the NN training process can improve the prediction accuracy of the trained model during the inference process. We build a self-built dataset of international students based on the existing dataset and train the model together. The international student dataset contains four emotions: happy, neutral, sad and angry, and using this dataset is more suitable for the primary application scenario in this study. Furthermore, we obtain a data pre-processing scheme that can be run in edge devices, which further improves the acceleration performance of our algorithm.

To prove the effectiveness of SFEARA algorithm, we added SFEARA algorithm to AlexNet, ResNet, GoogLeNet, and VGGNet for real performance comparison experiments. To obtain the theoretical performance improvement and energy consumption reduction, we performed simulation experiments on computational volume and memory access. The experiments demonstrate that our method is correlated with the dataset itself and can reduce energy consumption by 30–40%, performance improvement of 1.3×–1.6×, and accuracy reduced only within 3.3% in the dataset used in this study. Therefore, the present method is more suitable for images with more background information and has limited performance improvement in the face of images with less background information, which is a limitation of the SFEARA algorithm; however, at the overall network optimization level, the present method can be used for both cases, only the acceleration in the face of the second case is less effective. The model trained in this study can be used as a dedicated model for emotion recognition of international students, which has improved accuracy than the generic model and can provide an excellent model basis for related applications.

### Funding
This work was supported by the Education Research of Philosophy and Social Science Foundation (JJ194000) and the University Ideological and Polite Work Quality Improvement Foundation (23F08) of Hunan Province. The funders had no role in study design, data collection and analysis, decision to publish, or preparation of the manuscript.

### Grant Disclosures
The following grant information was disclosed by the authors:
Education Research of Philosophy and Social Science Foundation: JJ194000.
University Ideological and Polite Work Quality Improvement Foundation of Hunan Province: 23F08.

### Competing Interests
The authors declare that they have no competing interests.

### Author Contributions
- Lian Tong conceived and designed the experiments, analyzed the data, performed the computation work, authored or reviewed drafts of the article, and approved the final draft.
- Lan Yang conceived and designed the experiments, performed the experiments, performed the computation work, prepared figures and/or tables, and approved the final draft.
- Xuan Wang performed the experiments, analyzed the data, prepared figures and/or tables, and approved the final draft.
- Li Liu conceived and designed the experiments, performed the computation work, authored or reviewed drafts of the article, funding and Foreign Affairs Supporting, and approved the final draft.

### Data Availability
The code is available at Zenodo: JustLoveU, & Dou Ren. (2023). JustLoveU/SFEARA: v0.1.0 (v0.1.0). Zenodo. https://doi.org/10.5281/zenodo.7855148.

The datasets of MIRFLICKR-25K and MIRFLICKR-1M is available at The MIRFLICKR Retrieval Evaluation: https://press.liacs.nl/mirflickr/.

The Fer2013 dataset is available at Kaggle (main body and references): https://www.kaggle.com/datasets/msambare/fer2013.

### Supplemental Information
Supplemental information for this article can be found online at http://dx.doi.org/10.7717/peerj-cs.1611#supplemental-information.

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
