# Peer review of "Self-aware face emotion accelerated recognition algorithm: a novel neural network acceleration algorithm of emotion recognition for international students"

_PeerJ Computer Science, doi:10.7717/peerj-cs.1611_

## Round 0.1 · original submission · Major Revisions

Please carefully address the reviewers' comments, focusing especially on better clarifying the experimental scenario and parameters used, as well as better highlighting the original contribution.

Reviewer 1 ·

Basic reporting

The manuscript entitled “Self-aware face emotion accelerated recognition algorithm: A novel neural network acceleration algorithm of emotion recognition for international students” has been investigated in detail. The topic addressed in the manuscript is potentially interesting and the manuscript contains some practical meanings, however, there are some issues which should be addressed by the authors:
1) In the first place, I would encourage the authors to extend the abstract more with the key results. As it is, the abstract is a little thin and does not quite convey the interesting results that follow in the main paper. The "Abstract" section can be made much more impressive by highlighting your contributions. The contribution of the study should be explained simply and clearly.
2) The readability and presentation of the study should be further improved. The paper suffers from language problems.
3) The “Introduction” section needs a major revision in terms of providing more accurate and informative literature review and the pros and cons of the available approaches and how the proposed method is different comparatively. Also, the motivation and contribution should be stated more clearly.
4) The importance of the design carried out in this manuscript can be explained better than other important studies published in this field. I recommend the authors to review other recently developed works.
5) “Discussion” section should be added in a more highlighting, argumentative way. The author should analysis the reason why the tested results is achieved.
6) The authors should clearly emphasize the contribution of the study. Please note that the up-to-date of references will contribute to the up-to-date of your manuscript. The studies named- Artificial intelligence-based robust hybrid algorithm design and implementation for real-time detection of plant diseases in agricultural environments; Detection of solder paste defects with an optimization‐based deep learning model using image processing techniques- can be used to explain the method in the study or to indicate the contribution in the “Introduction” section.
7) The advantages of the proposed method compared with the others for the same class of problems in recent literature are not elaborated adequately. It is recommended to provide to simulation results compared with the others, which will show the performance of the proposed method in a clear manner.
8) The complexity of the proposed model and the model parameter uncertainty are not enough mentioned.
9) How to set the parameters of proposed method for better performance?
10) The effect of the parametric uncertainty is not discussed in detail. How did the comparison methods perform with or without the uncertainty?
11) It will be helpful to the readers if some discussions about insight of the main results are added as Remarks.
This study may be proposed for publication if it is addressed in the specified problems.

Experimental design

No comment.

Validity of the findings

No comment.

Additional comments

No comment.

·

Basic reporting

I have been reading your manuscript, and I have several questions and recommendations.

Table 5 would be more helpful if it includes a column with the number of images, as it is more or less the weight of each dataset.

Content of lines 473 to 475 (or a summarized version) should also appear as the Table 6 legend.

Line 533. A missing "E" in Experiments word

Figure 6 charts have the x axis mislabelled, as they should be "False Positive Rate" instead of "Fault Positive Rate"

S1 Table, Sheet 2: row 2 contents were not translated to English

Experimental design

Many human face expressions and hand movement gestures are learnt from the environment where the person was grown. Many cultural clashes can happen due misreadings of them.

What is an abnormal emotion from your point of view? Is that concept already embedded in your designs as an additional annotation? This question arises from the sentence in the introduction from line 43 to line 45.

Taking into consideration sentence from line 55 to 57, did you consider using subsets of Instagram, TikTok or Twitch as an additional source of training or testing materials? Their transient reels and videos are an usual tool for some feel expression (users feel more comfortable when what they have broadcasted is not always available).


As your image analysis methods are focused on identifying subtle facial changes, like facial muscles, which are the needed minimal conditions of the face (resolution, luminosity, head turn, skin tone, ornaments like piercings or make-ups adding, facial hair, pupil dilatation and eye color) to lower the false positives under an acceptable threshold? Did you have to generate separated models for different sets of populations or facial features in order to train it properly?

I have several questions about the ISSM_DataSet. Are the images dataset and its metadata deposited in an accessible repository for academic use? How many kind of different cultures are represented in the dataset? Which is their distribution? How many of the subjects were men? And women? Did you introduce as separate parameters to train the CNN sex and gender of the subjects?

At the paragraph starting at line 172 you are mentioning that AlexNet, GoogLeNet, VGGNet and ResNet networks were tested using ImageNet. Did you try the same with your developments? Is ImageNet an out of scope dataset for your developments? Why?

I guess that you downsampled MIRFLICK-1M. among other criteria, due many of the images did not have curated annotations. Were the criteria to downsample MIRFLICKR-25K through randomization the availability of sentiment annotations, or did you use additional criteria like avoiding biases in the training and test sets?

As MIRFLICKR-25K is a subset of MIRFLICKR-1M, did you check chosen images from MIRFLICKR-1M were not some of the images already chosen images from MIRFLICKR-25K, in order to avoid biases?

About the quality of the training materials, when you consolidated the datasets (MIRFLICKR-25K MIRFLICKR-1M and Fer2013 subsets) under the four possible sentiments, did you validate that the sentiment annotations from the different source datasets were really equivalent? Did you include images where more than one person appears? If it is so, how did you disambiguate which sentiment annotations had to to be assigned to which person in the image?

Validity of the findings

Have you gathered some additional metrics in order to cross-validate the power consumption reduction? For instance, peak memory consumption both at the CPU and the GPU levels, CPU and GPU times, etc...

Which are the CPU and memory consumptions of the pre-processing step of images? Is this proportional to the images resolution?

Additional comments

About SFEARA implementation, you stated at paragraph starting at line 435 the versions of Python and TensorFlow you used. Why did you use a discontinued version of Python and a 5 years old version of TensorFlow?

Where is the source code available of SFEARA implementation you have used for this manuscript's elaboration? Are you considering to also provide it in a software container in order to ease its usage when some other researcher willing to reproduce your work?

There are several typos along the manuscript. First of all, acronyms of GoogLeNet and VGGNet have some cases swapped.

---

## Round 0.2 · Minor Revisions

Please carefully address the comments of the Reviewers before submitting the revised version.

Reviewer 1 ·

Basic reporting

All my comments have been thoroughly addressed. It is acceptable in the present form.

Experimental design

Research question well defined, relevant & meaningful. It is stated how research fills an identified knowledge gap.

Validity of the findings

Conclusions are well stated, linked to original research question & limited to supporting results.

Additional comments

All my comments have been thoroughly addressed. It is acceptable in the present form.

·

Basic reporting

In general, you should be much more accurate in the details provided for the references in the references section, as some of them are incomplete or miss proper details, like some of the authors or the publisher.


Page 2, lines 63 and 73. FER acronym is used for the first time in those places, with no previous unambiguous introduction. Although it can be understood, FER acronym should have been introduced in line 63 or before, like "... in FER (facial emotion recognition) ..." or "... in facial emotion recognition (FER) ..."

Page 2, line 69. Should "existing public benchmark" be plural? i.e. "existing public benchmarks"

Page 2, line 71. The fragment of sentence "... the past one or two years ..." should explicitly mention the year instead of the relative temporal timeframe, so the manuscript's body understanding is not hindered by when it is being read.

Page 2, line 75. Sentence "However, the load consumption of various processing methods varies", is referring to the computational resources (memory, CPU, I/O) needed to achieve the task or to the energy needed to achieve it?

Page 2, line 98 and page 17, lines 537-538. I realized there was a typo at the reference "Khorasan et al. (Khorasani, 2001)". When I went to the references block in order to disambiguate it, I realized the reference from Khorasani had issues, due title repetition. It took me a while to find its details, but they are at https://spie.org/Publications/Proceedings/Paper/10.1117/12.436980?origin_id=x4323&start_year=2001 . Then I realized the list of authors was wrong, as it has two authors instead of one, being the first author Liying Ma, not Khashayar Khorasani. So the reference entry should be properly rewritten, and the mentions at lines 89 and 98 should be properly fixed.

Page 3, line 102 and line 105. Maybe " studies (Kyperountas et al., 2010; Zheng et al., 2006; Fu and Wei, 2008) " should be rewritten as " Kyperountas, Zheng and Fu studies (Kyperountas et al., 2010; Zheng et al., 2006; Fu and Wei, 2008)". Something similar happens on line 105.

Page 3, line 136. Should "... extracts feature from images ..." be "... extracts features from images ..."?

Page 3, line 144. CNN acronym is introduced here, but it has been used several times before this point is reached. My recommendation is to introduce it earlier.

Page 3, line 147, page 12, lines 422 and 426, and figure 7 description, page 13, line 435, page 14, table 5, page 16, line 498, and supplemental materials. "GoogleNet" should be written as "GoogLeNet" (for instance, see https://towardsdatascience.com/deep-learning-googlenet-explained-de8861c82765).

Page 3, line 151. Acronym "MLP", does it mean "Multilayer perceptron"?

Page 4, line 163-164. SVM acronym is introduced here, but it was already used in page 2. My recommendation is introducing it earlier.

Page 4, line 182. Maybe "tanh" and "relu" activation functions should be written has "hyperbolic tangent (tanh)" and "rectified linear unit (ReLU)" for the sake of clarity.

Page 5, block in line 192. Acronym "GD", does it mean "Gradient Descent", as it can be partially sensed from sentences at line 195 and onwards?

Page 8, table 3. I have been having a look at the numbers, and it seems there is some kind of outlier in the single image runtime for MIRFLICKR-1M and SSD-300, compared to using SSD-300 with the other datasets (an order of magnitude), or comparing the proportions of runtime in the other rows with the other programs. Is there some typo, or is it really an outlier?

Page 8, line 319. I do not understand the sentence starting "Then, combine the framework api and the model provided by the model in ...". Which "api", which framework?

Page 8, line 323. What does "LBP" mean? Local binary patterns? This acronym should be described.

Page 10, line 381. What does "ROI" mean? I guess it is not "return of investment". Is it "region of interest"? This acronym should be declared.

Page 17, lines 565-566. The list of authors of this reference is incomplete, as it lacks a fourth and last author, from what I learnt from https://ieeexplore.ieee.org/document/6909614

Experimental design

No comment

Validity of the findings

No comment

---

## Round 0.3 · accepted · Accept

The Authors timely addressed the Reviewers' comments. The manuscript is now suitable for publication.